# The Mediating Role of Psychological Well-Being in the Relationship between Self-Care Knowledge and Disease Self-Management in Patients with Hypertensive Nephropathy

**DOI:** 10.3390/ijerph19148488

**Published:** 2022-07-12

**Authors:** Wen-Chun Chen, Shu-Fang Vivienne Wu, Juo-Hsiang Sun, Chun-Yi Tai, Mei-Chen Lee, Chun-Hua Chu

**Affiliations:** 1Department of Nursing, Chiayi Campus, Chang Gung University of Science and Technology, Taoyuan City 600, Taiwan; shermie@stm.org.tw; 2Department of Community Medicine, St. Martin De Porres Hospital, Chiayi 600, Taiwan; 3School of Nursing, National Taipei University of Nursing and Health Sciences, Taipei 112, Taiwan; shufang@ntunhs.edu.tw (S.-F.V.W.); yii@ntunhs.edu.tw (C.-Y.T.); 4Department of Long-Term Care, National Taipei University of Nursing and Health Sciences, Taipei 112, Taiwan; juohsaingsun@gmail.com; 5Research Center for Healthcare Industry Innovation, National Taipei University of Nursing and Health Sciences, Taipei 112, Taiwan; 8chunhua@ntunhs.edu.tw

**Keywords:** hypertensive nephropathy, psychological well-being, self-care knowledge, disease self-management, mediating role

## Abstract

Objective: This study aimed to investigate the correlation between self-care knowledge, psychological well-being, and disease self-management in patients with hypertensive nephropathy, and to assess the effect of psychological well-being as a mediator of self-care knowledge and disease self-management. Methods: This is a cross-sectional study. The 220 patients with hypertensive nephropathy were recruited from a teaching hospital in Taiwan using purposive sampling. The average age was 70.14 (SD = 11.96) years old. Among them, 128 (58.2%) were male and 92 (41.8%) were female. Instruments included a hypertensive nephropathy self-care knowledge scale, the World Health Organization-5 Well-Being Index, and the chronic kidney disease self-management instrument. The mediating effect was determined with linear regression models and the Sobel test. Results: The total explanatory variation of age, systolic blood pressure, psychological well-being, and self-care knowledge on the disease self-management was 27.7%. Psychological well-being was the most important explanatory factor and alone explains 16%. Psychological well-being was a partial mediator of self-care knowledge and quality of life in patients with hypertensive nephropathy, with a total effect of 23.2%. Conclusions: This study showed that older patients with hypertensive nephropathy and those with a higher systolic blood pressure had lower levels of disease self-management. The higher the patients’ self-care knowledge and psychological well-being, the better their disease self-management.

## 1. Introduction

The 2019 Global Burden of Disease data collected by the Institute for Health Metrics and Evaluation at the University of Washington showed that disability-adjusted life years increased from 27.8% in 1990 to 66% in 2019 due to noninfectious chronic diseases [1]. According to the 2016 global burden of hypertension data from the World Hypertension League, the number of deaths caused by hypertension was 23,901 in Taiwan, which accounted for 14.6% of total deaths. Additionally, the number of people with disabilities attributable to hypertension was 446,072, accounting for 7.9% of all people with disabilities [2]. Furthermore, inadequate control of hypertension may result in the development of hypertensive nephropathy, which is one of the causes of end-stage kidney disease (ESKD). Statistics from the United States Renal Data System revealed an ESKD incidence rate per million of 523 in Taiwan in 2018, which was the highest in the world. The prevalence rate per million was 2587, which was the second highest globally and the highest among Asian countries [3]. Risk factors include smoking, diabetes, high cholesterol levels, obesity, lack of physical activity, and unhealthy diet. These factors increase the risk for cardiovascular disease in hypertensive patients [4]; previous studies have suggested that uncontrolled hypertension increases the risk of coronary artery disease, myocardial infarction, heart failure, stroke, and peripheral artery disease, as well as renal diseases such as kidney failure and comorbidities such as impaired vision [5,6,7].

Hypertensive nephropathy is a chronic disease consisting of a concurrent diagnosis of two chronic diseases. Therefore, the disease self-care knowledge for hypertension, chronic kidney disease (CKD), and hypertensive nephropathy differs slightly. The diet control, medication used, and daily life precautions for hypertensive nephropathy are different from those for patients with a single chronic disease. Moreover, CKD and other chronic diseases interact with each other, so chronic disease prevention should be combined with disease self-care information to improve efficacy and prevent the onset of comorbidities [8]. Controlling blood pressure is the most important thing for managing hypertensive nephropathy, which should include diet changes such as a low sodium diet, weight loss, smoking cessation, and regular exercise.

Self-care at home entails going to follow-ups at clinics, controlling the disease or factors causing kidney function deterioration, such as the “three highs” (hypertension, high cholesterol, and high blood sugar) and other chronic diseases, not taking painkillers or medications of an unknown origin, preventing infections, using public health resources, and staying in touch with a renal disease medical team. If these changes are not able to control the hypertension, blood pressure medication is required [9,10]. However, angiotensin-converting enzyme inhibitors and angiotensin receptor blockers should be avoided when choosing blood pressure medication, especially for patients with early-stage CKD or pre-ESKD. Patients should be attentive to nephrotoxic side effects to avoid progression into ESKD. Moreover, each patient’s compliance with medication should be monitored, regular follow-ups should be advised, and medication should be adjusted accordingly by doctors [11,12]. Self-care knowledge includes going to regular follow-ups, monitoring blood test results, fostering disease self-management and health responsibilities (e.g., understanding normal blood pressure and self-assessment procedures), maintaining a healthy diet, developing a healthy lifestyle (e.g., quitting smoking and drinking and regular exercise), having a positive attitude and coexisting with the disease, increasing social interactions and support, and managing stress [13,14].

Mental health and psychological well-being (PWB) are very related concepts. Individuals can set their own standard by assessing their current quality of life and developing a more stable awareness and emotional state [15]. Mental health can be differentiated into subjective well-being and PWB. These two concepts differ in their explanation of happiness [16]. The former is more subjective, and defines happiness as feelings of contentment; it is an overview of one’s life satisfaction and emotional status (positive or negative). PWB perceives happiness as an objective standard, accentuates self-fulfillment, and indicates rational and positive dimensions [17]. Ryff [18], a psychologist in the United States, proposed a model of PWB with six factors: self-acceptance, positive relations with others, autonomy, environmental mastery, purpose in life, and personal growth. Other studies have suggested that PWB and comfort both implicate happiness [19].

Hypertensive nephropathy is a type of chronic illness that may cause mental distress, and is often comorbid with depression [20]; the occurrence of this comorbidity in chronic disease patients is more than twice compared to healthy individuals [21]. In Europe and the Americas, mental distress is a major cause of disability. Research suggests that the depression rate is 30–40% in chronic disease patients [22]. One study reported that 14.1% of hypertension patients have symptoms of depression [23]. When hypertension is inadequately controlled, prolonged illness may cause depressive emotions that affect PWB and jeopardize happiness [24]. Therefore, clinical care providers need to screen for depressive symptoms in hypertensive nephropathy patients and deliver interventions early on to help patients maintain a good quality of life.

For patients with hypertensive nephropathy, facing being diagnosed with an incurable disease, coping with the challenges of chronic illness, coexisting with the disease long-term, and demonstrating good disease self-management are important. Chuang et al. [25] demonstrated that self-care knowledge, self-efficacy, and self-management are positively correlated in patients with early-stage CKD. Self-efficacy was an important single explanatory variable for predicting self-management, and explained 49% of the variation. Moreover, a study that focused on patients with stage 1 to 5 CKD found that disease knowledge, self-efficacy, and self-care behaviors are positively associated. Patients’ disease knowledge predicts self-care behaviors via self-efficacy, which was found to act as a complete mediating variable with a total effect of 50% [26]. Another study assessed the mediating effect of happiness on social support, family function, and self-management in older patients with hypertension. The results showed that social support, family function, happiness, and self-management were positively correlated. Using a stepwise multiple regression analysis, it was found that social support and family function were not important predicting factors for self-management. Happiness was a partial mediating variable of social support and self-management (*p* < 0.05) and explained 89.9% of the variance; the R^2^ value was 66.3%. Studies have suggested that patients can achieve self-management by improving their social support, family function, and increasing their happiness [27].

For medical staff, facilitating chronic disease patients’ long-term care and reducing the chance of comorbidity are big challenges. A lack of self-care knowledge may cause mental distress or even depression, which in turn affects patients’ self-management abilities. Moreover, uncontrolled hypertension can trigger comorbidities and severely compromise quality of life. Current research mainly focuses on patients with hypertension or CDK. Further investigation is required to determine whether there is a correlation between self-care knowledge, PWB, and disease self-management, as well as whether self-care knowledge affects self-management and whether patients’ self-management is affected by improving PWB. The purpose of this study was, therefore, to identify the correlations between self-care knowledge, PWB, and disease self-management in patients with hypertensive nephropathy and determine the effects of PWB as a mediating factor for self-care knowledge and disease self-management.

## 2. Methods

### 2.1. Setting and Participants

A cross-sectional correlation study design was used. A total of 220 patients were recruited from an outpatient nephrology clinic at a regional teaching hospital in Taiwan using purposive sampling. Inclusion criteria were: (1) over 20 years old; (2) clear consciousness; (3) no prior mental illness diagnosis; (4) ability to communicate in Mandarin or Taiwanese; and (5) diagnosed with hypertensive nephropathy or pre-ESKD (stage 3b to stage 5 CKD) by a physician. Exclusion criteria included: (1) CKD stage 5 patients who had received dialysis treatment; (2) patients diagnosed with diabetic nephropathy by a physician; (3) cognitive dysfunction and inability to understand the questionnaire; and (4) medical history of mental illness (e.g., schizophrenia). This study was reported in compliance with the Strengthening the Reporting of Observational Studies in Epidemiology (STROBE) recommendations (Appendix A).

### 2.2. Measures

#### 2.2.1. Demographic Data and Disease Characteristics

Demographic information included age, gender, education level, marital status, previous medical history, smoking or drinking habits, living conditions, whether the patient had received public health education about hypertension, and whether the patient monitors their blood pressure regularly. Disease characteristics comprised CKD stage and the number of years the patient had had hypertension or CDK. Physiology parameters were systolic blood pressure (SBP), diastolic blood pressure, estimated glomerular filtration rate (eGFR), total cholesterol, and triglyceride levels.

*Hypertensive Nephropathy**Self-Care Knowledge Scale (HN-SCK**)*: This scale was derived from Chen et al. [28]. It is based on the ESKD patient care public health education from the Taiwan Ministry of Health and Welfare [29] and previous literature on hypertension and CDK care. There are three major topics related to daily self-care: (1) disease knowledge; (2) diet control; and (3) daily life precautions. There are a total of 12 items in the HN-SCK scale. A two-point dichotomous scale was used, with answers of yes, no, or I do not know. Answering yes corresponded to 1 point. The answer of “I do not know” equals “No”, and had no points. The total scores ranged from 0 to 12. All items were positively keyed and no negatively keyed questions were asked. The higher the total score, the better the self-care knowledge. Explanatory factor analysis was used to test the reliability and validity of the questionnaire. The factor loading of the three subscales was 0.44–0.90 and the variance explained was 57.65%. The average content validity index was 0.98 and the internal consistency reliability had a Cronbach’s α of 0.81 [28]. In this study, the KR-20 reliability of this scale was 0.82.

#### 2.2.2. World Health Organization-5 Well-Being Index (WHO-5)

WHO-5 is a short self-reported measure of current PWB. The measure was first presented in 1998 by the WHO Regional Office in Europe as part of the DEPCARE project on well-being measures in primary health care. It has been published in more than 30 national languages so far, as a tool to screen people for depression, offering good reliability and validity [30]. A 6-point scale was adopted, ranging from 0 to 5 points, 0 being never (worst) and 5 being all the time (best). The total scores ranged from 0 to 25 points. If a patient has a total score less than 13 or answers any items corresponding to 0 or 1 point, a severe depression scale was advised, or a psychiatrist was required to assess whether the patient had depression. The internal consistency reliability had a Cronbach’s α of 0.87, and the retest reliability had a Cronbach’s α of 0.90 in Japan and Germany [31]. This study used this scale to assess the PWB of patients with hypertensive nephropathy. The Cronbach’s α value for this study was 0.89.

#### 2.2.3. Chronic Kidney Disease Self-Management Instrument (CKD-SM)

The CKD-SM was developed by Lin et al. [32] for patients with early-stage CKD. This scale comprises 29 items in total, including four subscales: problem-solving (10 items), self-integration (10 items), seeking social support (5 items), and compliance behavior (4 items). This instrument used a 4-point Likert scale for scoring (1 = never, 2 = sometimes, 3 = often, and 4 = always), with total scores ranging from 1 to 116 points. The higher the total score, the better the patients’ self-management. This study performed exploratory factor analysis to test the reliability and validity of results, and adopted the Kaiser–Meyer–Olkin value of 0.92. The factor loading of the four subscales was 0.57–0.87 and the variance explained was 60.51%. The internal consistency reliability of the overall scale had a Cronbach’s α value of 0.77–0.92, and the two-week test–retest reliability was 0.72 [32]. In Taiwan, this scale has been used in many studies related to CKD. This scale has good reliability and validity, and can be used as an instrument for evaluating the disease self-management of patients with hypertensive nephropathy. The content validity index was 0.96, and the Cronbach’s α value for this study was 0.84.

### 2.3. Data Collection

Data were collected from April 2020 to March 2021. Once the participants met the inclusion criteria, after being introduced by the attending physician of the department of nephrology and receiving informed consent, the investigators conducted the survey. The questionnaire comprised four sections: self-care knowledge, PWB, disease self-management, and patient demographic and disease characteristics. Before the investigation, all investigators were trained on the inclusion and exclusion criteria, communication skills, and data-collection strategies.

### 2.4. Statistical Analysis

The sample size of this study was calculated by the statistical software G-power [33]. α was set to be 0.05 and the power was set to be 0.95. We referred to a previous study and set the effect size to be 0.30 [28]. The estimated sample size was at least 134 people. Ultimately, we recruited 220 patients for this study. SPSS 20.0 (released 2011; IBM Corporation, Armonk, NY, USA) was used for documentation and statistical analysis following the recruitment process. The mean, standard deviation (SD), frequency distribution, and percentage were used for descriptive statistics including demographics, disease characteristics, and the distributions for each scale. Pearson’s correlation coefficient was used to assess the correlations between the variables. Variables that were significantly different and showed correlations were analyzed further through stepwise regression analysis to test their explanatory power for disease self-management. To determine the mediating effect of PWB, we used the three regression equations proposed by Baron and Kenny [34]. We conducted the analysis with a linear regression, and used Sobel’s test to assess the variables’ mediating effects on disease self-management.

### 2.5. Ethical Considerations

This study was approved by the Institutional Review Board of the recruitment hospital upon initiation of the investigation (IRB number: CTH-108-3-1-016). All participants entered the study voluntarily and signed a consent form after being given an explanation of the research. The data were compiled and analyzed anonymously. The research data were properly protected and were used for academic research purposes only. We guaranteed that the study would not harm the rights of the participants and they were told dropping out of the study would not incur any harm.

## 3. Results

### 3.1. Sample Characteristics

The ages of the participants ranged between 31 to 99 years old. The mean age was 70.14 years old (SD = 11.96). The majority were men (*n* = 128, 58.2%) and had a high school level of education (*n* = 82, 37.3%). The mean number of years that patients had been diagnosed with hypertension was 15.36 (SD = 10.15) and the mean number of years being diagnosed with hypertensive nephropathy was 5.64 (SD = 5.82). The mean self-care knowledge, PWB, and disease self-management scores of patients were 9.35 (SD = 2.46), 18.38 (SD = 5.16) and 81.69 (SD = 16.13) points, respectively. The remaining patient demographics and disease characteristics are shown in Table 1.

### 3.2. Correlation Analysis of Demographics, Disease Characteristics, Self-Care Knowledge, Psychological Well-Being, and Disease Self-Management

We discovered that disease self-management was positively correlated with eGFR (*r* = 0.263, *p* < 0.05), self-care knowledge (*r* = 0.374, *p* < 0.01), and PWB (*r* = 0.461, *p* < 0.01). This indicated that the better the patients’ kidney function, the higher their self-care knowledge, PWB, and disease self-management. The age of participants (*r* = −0.185, *p* < 0.05) and their SBP (*r* = −0.161, *p* < 0.05) were negatively correlated with disease self-management, suggesting that the older the patient, the higher their SBP, and the worse their disease self-management. Moreover, the self-care knowledge and PWB of patients were positively associated (*r* = 0.254, *p* < 0.01), indicating that PWB increased with higher self-care knowledge scores. Additionally, age and SBP were shown to be negatively correlated with self-care knowledge (age: *r* = −0.139, *p* < 0.05; SBP: *r* = −0.142, *p* < 0.05). This indicates that old age and high SBP were associated with lower self-care knowledge (Table 2).

### 3.3. Predicting Variables for Disease Self-Management

We found that disease self-management was significantly correlated with age, SBP, PWB, and self-care knowledge in patients with hypertensive nephropathy using a correlation analysis. To assess the effect and explanatory power of the variables mentioned on disease self-management, a stepwise regression analysis was carried out. It was found that the four variables, age, SBP, self-care knowledge, and PWB, explained 27.7% of the total variance of disease self-management (R^2^ = 0.277, *p* < 0.001). Among them, PWB was the most important single explanatory variable and explained 16% of the variance alone (R^2^ change = 0.160, *p* < 0.001) (Table 3).

### 3.4. The Mediating Role of Psychological Well-Being on Self-Care Knowledge and Disease Self-Management

As mentioned above, self-care knowledge and disease self-management are correlated. The lower the patients’ self-care knowledge, the worse their disease self-management. To predict whether interventions targeting PWB can improve the disease self-management of patients with low self-care knowledge, the mediating effect was determined in two steps. In step one, a linear regression analysis was used. First, an equation was used to test the regression relationship between the self-care knowledge explanatory variable and the mediator, PWB. We identified that self-care knowledge and PWB differed significantly (R^2^ = 0.07, *p* < 0.001). The second equation determined the regression relationship of the self-care knowledge explanatory variable and the outcome variable, disease self-management. Self-care knowledge and disease self-management were statistically different (R^2^ = 0.14, *p* < 0.001). The last equation inputted the explanatory variable, self-care knowledge, and the mediator in a regression model to understand the effects the explanatory variable (self-care knowledge) had on the outcome variable (disease self-management) after controlling for the mediator. The results showed that when controlling for PWB, the correlation between the explanatory variable and outcome variable differed significantly (R^2^ = 0.24, *p* < 0.001). A partial mediation effect was thus observed (Figure 1).

The second step was to use a Sobel test, as proposed by Sobel [35]. From the linear regression analysis, we learned that the unstandardized regression coefficient was 0.53 and the standard error was 0.14 between the explanatory variable and the mediator. The unstandardized regression coefficient was 1.021 and the standard error was 0.191 between the mediator and the outcome variable. The Z-value was 3.13, as calculated by the Sobel test, and the standard error was 0.174 (*p* < 0.001). Therefore, PWB was determined to exert a partial mediating effect on self-care knowledge and disease self-management; the total effect was 23.2% (Table 4).

## 4. Discussion

The correct response rate of the self-care knowledge questionnaire was above average (77.92%) in hypertensive nephropathy patients, which is higher than that reported in Ethiopia (68.7%; [36]), Indonesia (50%; [37]), and Malaysia (26.3%; [38]). The discrepancy in the correct response rates can be attributable to Taiwan’s public health policy for promoting mass screening in their healthcare integration program. This exposed participants to relevant health education in internal cardiovascular and family doctor clinics early on during their hypertensive nephropathy diagnosis. Moreover, the recruited participants were patients with pre-ESKD, which is covered by the universal healthcare plan, and the recruitment hospital was one of the participating health education facilities. The self-care knowledge of participants in this study was thus above average.

We found that age and disease self-management were negatively correlated, with a higher age worsening patient disease self-management. This result differs from a study conducted by Wu et al. [26] that found that age and self-care behaviors were positively associated in CKD patients, meaning that the older the patient, the better their self-care behaviors. This inconsistency may be due to the participating population, and the fact that the CDK stages of the participants were not differentiated. In addition, the more time that passes since diagnosis with the disease, the older the patients are; these participants are provided with more health guidance from their medical team, which results in better self-care behaviors. Furthermore, the four variables of age, SBP, self-care knowledge, and PWB are predicting variables of disease self-management, with PWB as the most important explanatory variable. This result is consistent with Chuang et al. [25], who found that self-care knowledge and self-management were positively correlated in early CDK patients.

We investigated whether PWB mediates self-care knowledge and disease self-management in hypertensive nephropathy patients. We found that PWB was a partial mediator on self-care knowledge and disease self-management, but not completely. This is similar to the findings of Zhang et al. [27], where happiness was found to be a partial mediator of family function and self-management in older hypertension patients. Our study identified self-care knowledge as a predictor of disease self-management in patients with hypertensive nephropathy, and suggests that by improving PWB, disease self-management can be enhanced. However, we noticed that the R^2^ of 16% was relatively low for PWB as a single predictor variable of disease self-management in patients with hypertensive nephropathy using a stepwise regression analysis. Moreover, self-care knowledge as a single predictor only explained 7.9% of the variance. Therefore, PWB is a partial mediating variable.

### Limitations

A cross-sectional study design was used, so the investigation was done at a certain period in time. The correlations between the variables from the 220 participating hypertensive nephropathy patients are thus applicable only in this certain timeframe. We recommend expanding the participant population and recruiting from medical facilities with all levels of care because the results may differ. In addition, we recommend implementing an intervention to improve disease self-management to observe the progression and effects, and potentially deduce causal relationships between variables. Furthermore, mediators other than values using Sobel’s test can also enter the confidence intervals and utilize the Bootstrap technique. The result is perhaps more precise.

## 5. Conclusions

In conclusion, we discovered that PWB and self-care knowledge correlated significantly with disease self-management in patients with hypertensive nephropathy. The PWB of patients was the single most important explanatory variable of disease self-management. Therefore, increasing the PWB and happiness of patients can in turn encourage disease self-management. We advise that nursing clinicians further assess the parameters of age, SBP, PWB, and self-care knowledge. Older patients or patients with a high SBP should be given personal health guidance to increase their self-care knowledge and promote disease self-management. We showed that self-care knowledge and PWB were positively correlated. PWB was a partial mediator for improving disease self-management. We, therefore, recommend promoting self-care knowledge education, health guidance, and public health interventions by nursing clinicians to foster disease self-management. Furthermore, we recommend that nursing clinicians screen for patients in mental distress. Patients with mental health problems or low happiness levels require prompt medical attention to promote disease self-management and to help them make peace with their hypertensive nephropathy disease.

We found that PWB exerts a partial mediating effect on self-care knowledge and disease self-management. The self-care knowledge of patients with hypertensive nephropathy predicts disease self-management. Increasing PWB can thus promote disease self-management. Furthermore, we found that age, SBP, self-care knowledge, and PWB were predicting variables for disease self-management, with PWB being the most important explanatory variable. Therefore, we advise that nursing clinicians assess the parameters of age, SBP, self-care knowledge, and PWB. In addition, reducing mental distress and promoting PWB or happiness by encouraging patients to join a support group or providing social resources may improve disease self-management. This can eventually allow patients with hypertensive nephropathy to live in balance with their chronic illness. This study thus lays the ground for future research on nursing interventions.

## Figures and Tables

**Figure 1 ijerph-19-08488-f001:**
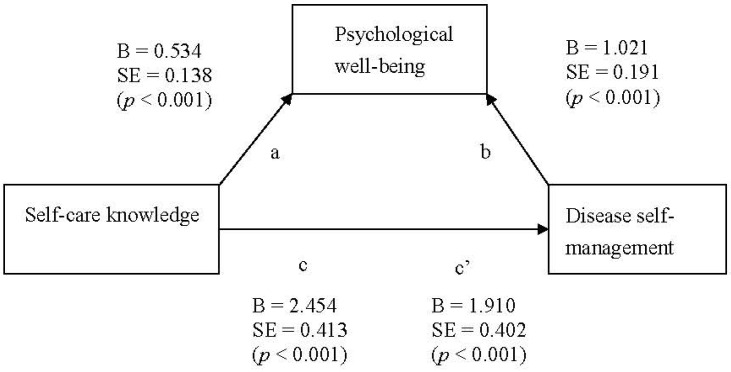
Mediating effect of psychological well-being.

**Table 1 ijerph-19-08488-t001:** Demographic data and disease characteristics of the patients with hypertensive nephrology (*n* = 220).

Variables	Min	Max	Mean	SD
Age (years)	31	99	70.14	11.96
Duration of hypertension (years)	0	50	15.36	10.15
Duration of hypertension nephropathy (years)	0	30	5.64	5.82
Systolic blood pressure (mmHg)	101	187	135.62	14.69
Diastolic blood pressure (mmHg)	46	98	71.81	9.73
estimated Glomerular Filtration Rate (mL/min/1.73 m^2^)	6.1	43.6	25.77	13.88
Total cholesterol (mg/dL)	87	260	163.34	33.22
Triglyceride (mg/dL)	36	108	139.33	100.18
Low-density lipoprotein (mg/dL)	1	193	92.20	30.00
Charlson comorbidity index	1	7	2.81	1.41
Self-care knowledge	2	12	9.35	2.46
Psychological well-being	5	25	18.38	5.16
Disease self-management	36	116	81.69	16.13
	** *n* **	**%**		
Gender				
Male	128	58.2		
Female	92	41.8		
Religious belief				
Buddhism	53	24.1		
Taoism	104	47.3		
Christianity/Catholic	20	9.1		
Other	43	19.5		
Educational level				
Illiteracy	14	6.4		
Elementary school	53	24.1		
Junior high school	34	15.5		
High school	82	37.3		
Above university	37	16.8		
Marital status				
Unmarried	9	4.1		
Married	180	81.8		
Divorced or widowed	31	14.1		
Employment				
No	167	75.9		
Yes	53	24.1		
Smoking				
No	199	90.5		
Yes	21	9.5		
Drinking				
No	205	93.2		
Yes	15	6.8		
Exercise				
No	144	65.5		
Yes	76	34.5		

Abbreviations: SD, standard deviation; Min, Minimum; Max, Maximum.

**Table 2 ijerph-19-08488-t002:** Analysis of the correlation among demographic data, physiological index, self-care knowledge, psychological well-being, and disease self-management in patients with hypertensive nephropathy (*n* = 220).

Variables	Age	CCI	Duration of HT	Duration of HN	SBP	DBP	eGFR	TCH	TG	LDL	PWB	SCK	DSM
Age (years)	1												
CCI	0.041	1											
Duration of HT (years)	0.428 **	0.197 **	1										
Duration of HN (years)	0.254 **	0.059	0.293**	1									
SBP (mmHg)	0.192 **	0.009	0.205 **	0.040	1								
DBP (mmHg)	−0.146 *	−0.051	−0.137 *	−0.067	0.126	1							
eGFR (mL/min/1.73 m^2^)	−0.162 *	−0.206 **	−0.158 *	−0.238 **	−0.132	−0.026	1						
TCH (mg/dL)	−0.226 **	0.122	−0.069	0.041	0.025	0.012	−0.064	1					
TG (mg/dL)	−0.079	0.225 **	−0.006	−0.068	0.057	0.038	0.002	0.257 **	1				
LDL (mg/dL)	−0.149 *	0.090	−0.113	−0.071	−0.094	0.018	0.076	0.731 **	0.081	1			
PWB	0.030	−0.028	−0.032	−0.125	−0.021	−0.128	0.069	−0.017	−0.013	−0.072	1		
SCK	−0.139 *	−0.095	−0.042	0.017	−0.142 *	0.001	0.007	0.082	−0.001	0.009	0.254 **	1	
DSM	−0.185 *	−0.024	−0.086	0.016	−0.161 *	−0.011	0.263 *	0.114	0.005	0.046	0.461 **	0.374 **	1

Abbreviations: CCI, Charlson comorbidity index; HT, hypertension; HN, hypertensive nephropathy; SBP, systolic blood pressure; DBP, diastolic blood pressure; eGFR, estimated Glomerular Filtration Rate; TCH, total cholesterol; TG, triglyceride; LDL, low-density lipoprotein; PWB, psychological well-being; DSM, disease self-management; SCK, self-care knowledge; * *p* < 0.05; ** *p* < 0.01.

**Table 3 ijerph-19-08488-t003:** Stepwise regression analysis results of the disease self-management in patients with hypertensive nephropathy.

Variables	B	*β*	R^2^	R^2^ Change	F	*p*	95% CI
PWB	1.252	0.400	0.160	0.160	41.561	<0.001	0.869; 1.634
SCK	1.910	0.291	0.239	0.079	34.106	<0.001	1.117; 2.702
Age	−0.186	−0.138	0.258	0.019	24.991	<0.001	−0.343; −0.028
SBP	1.188	0.156	0.277	0.020	20.633	<0.001	0.221; 2.154

Abbreviations: PWB, psychological well-being; SCK, self-care knowledge; SBP, systolic blood pressure; CI, confidence intervals.

**Table 4 ijerph-19-08488-t004:** Linear regression analysis of mediating effect of psychological well-being between self-care knowledge and disease self-management.

IV	DV	*R* ^2^	Adjusted R^2^	*F*	Standard Error	Standardized β	*t*	*p*
Step1 SCK	PWB	0.065	0.060	15.044	0.534	0.138	3.879	<0.001
Step2 SCK	DSM	0.140	0.136	35.360	2.454	0.413	5.946	<0.001
Step3 SCK	DSM	0.239	0.232	34.106	1.910	0.402	4.748	<0.001
PWB					1.021	0.191	5.330	<0.001

Abbreviations: IV, independent variables; DV, dependent variables; SCK, self-care knowledge; PWB, psychological well-being; DSM, disease self-management.

## Data Availability

Due to the nature of this research, participants of this study did not agree for their data to be shared publicly, so supporting data are not available.

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
