# Peer review of "The Mediating Role of Psychological Well-Being in the Relationship between Self-Care Knowledge and Disease Self-Management in Patients with Hypertensive Nephropathy"

_ijerph, 2022, doi:10.3390/ijerph19148488_

Round 1
Reviewer 1 Report
Introduction - outline the topic supported by data from reports.
Line 133: In the methodology you write about random sampling, while the abstract mentions purposive sampling: ' The 220 patients with hypertensive nephropa-16 thy were recruited from a teaching hospital in Taiwan using purposive sampling.’.'
Line 156: You did not use a Likert scale only a nominal scale.
In simplified terms, a 'Likert scale' is a cafeteria of 5 statements arranged in order from total rejection to total acceptance, e.g.: strongly disagree, rather disagree, no opinion, rather agree, strongly agree.
Sometimes it is more elaborate, e.g. : absolutely not, no, I think not, I don't know, I think yes, yes, absolutely yes.
Line 353-354 and line 369-370: Two of the same recommendation.
Author Response
Point 1: Introduction - outline the topic supported by data from reports..
Response 1: Thank you for the comments.
Point 2: Line 133: In the methodology you write about random sampling, while the abstract mentions purposive sampling: ' The 220 patients with hypertensive nephropa-16 thy were recruited from a teaching hospital in Taiwan using purposive sampling.
Response 2: Thank you for the comments. Author has deleted “random sampling”. And change to “purposive sampling”.
Point 3: Line 156: You did not use a Likert scale only a nominal scale.
In simplified terms, a 'Likert scale' is a cafeteria of 5 statements arranged in order from total rejection to total acceptance, e.g.: strongly disagree, rather disagree, no opinion, rather agree, strongly agree.
Sometimes it is more elaborate, e.g. : absolutely not, no, I think not, I don't know, I think yes, yes, absolutely yes.
Response 3: Thank you for the comments.
Likert scale had deleted. The author use a dichotomous scale is a two-point scale that presents options that are absolutely opposite to each other. Yes or No or I don't know. Answers I don't know equals no, no points.
Point 4: Line 353-354 and line 369-370: Two of the same recommendation.
Response 4:Line 369-370 had deleted. Thank you for the comments.
Reviewer 2 Report
Thank you for the opportunity to review this study entitled “The Mediating Role of Psychological Well-Being in the Relationship between Self-Care Knowledge and Disease Self-Management in Patients with Hypertensive Nephropathy” (ijerph-1782014).
The study focused on the mental health of patients with hypertensive nephropathy, by exploring the relationship between self-care knowledge, psychological well-being, and disease self-management. A sample of 220 patients with hypertensive nephropathy was involved in the research.
In my opinion, the research topic is relevant, and the study is interesting. Parallelly, there are some issues that need to be addressed before the paper will be suitable for publication.
1. Abstract: the information about the sample should be deepened (Mean age and SD? Percentage of men and women?) to provide a clear picture of what will be presented in the paper.
2. Although in my opinion the section of the method is written in an excellent way, I believe that the information regarding the administration needs to be deepened: was it a one-to-one setting or a collective one? How long did the administration last? How were the missing values managed? Etc ...
3. Although I recognize the value of the Sobel test, in the mediation analysis it would be important to also enter the confidence intervals and use the Bootstrap technique.
In general, I really enjoyed this paper, which seems to be well structured, interesting, and pleasant to read. In my opinion, after the authors make small changes, it will be ready to be published.
Author Response
Response to Reviewer 2 Comments
Point 1: Thank you for the opportunity to review this study entitled “The Mediating Role of Psychological Well-Being in the Relationship between Self-Care Knowledge and Disease Self-Management in Patients with Hypertensive Nephropathy” (ijerph-1782014).
The study focused on the mental health of patients with hypertensive nephropathy, by exploring the relationship between self-care knowledge, psychological well-being, and disease self-management. A sample of 220 patients with hypertensive nephropathy was involved in the research.
In my opinion, the research topic is relevant, and the study is interesting. Parallelly, there are some issues that need to be addressed before the paper will be suitable for publication.
Response 1: Thank you so much for your affirmation.
Point 2: Abstract: the information about the sample should be deepened (Mean age and SD? Percentage of men and women?) to provide a clear picture of what will be presented in the paper.
Response 2: Thanks for the comment.
The author has added your suggestion in the Abstract. There were 220 participants in this study, with an average age of 70.14 (SD = 11.96) years old. Among them, 128 (58.2 %) were male, and 92 (41.8%) were female.
Point 3: Although in my opinion the section of the method is written in an excellent way, I believe that the information regarding the administration needs to be deepened: was it a one-to-one setting or a collective one? How long did the administration last? How were the missing values managed? Etc ...
Response 3: Thanks for the comment.
Collection is a one-to-one setup. A questionnaire, about 15-20 minutes, with no missing values.
Point 4: Although I recognize the value of the Sobel test, in the mediation analysis it would be important to also enter the confidence intervals and use the Bootstrap technique.
Response 4: Thanks for the comment. According to enter the confidence intervals and use the Bootstrap technique. Maybe the result is more precise.
The author has added in Limitation section.
Point 5: In general, I really enjoyed this paper, which seems to be well structured, interesting, and pleasant to read. In my opinion, after the authors make small changes, it will be ready to be published.
Response 5 :Thank you so much for your affirmation.